# Traditional medicine consumption in postpartum for HBV-infected women enrolled in the ANRS 12345 TA PROHM study in Cambodia

Sotheara Moeung[1]*, François Chassagne[2], Sophie Goyet[3], Sovann Nhoeung[4], Lynecta Sun[5], Dorina Yang[5], Steve Vilhem[6], Bunnet Dim[4], Socheat Ly[4], Linda Sov[4], Vouchleang Sreng[4], Sokda Chorn[4], Samsorphea Chhun[7], Laurence Borand[4], Sothea Kim[5], Olivier Segeral[8,9]

1 GMO, University of Health Sciences, Phnom Penh, Cambodia, 2 UMR 152 PharmaDev, Université de Toulouse, IRD, UPS, Nice, France, 3 "Independent Researcher", 7 Passage du Clair Matin, Annecy le Vieux, France, 4 Epidemiology and Public Health Unit, Institut Pasteur du Cambodge, Phnom Penh, Cambodia, 5 Faculty of Pharmacy, University of Health Sciences, Phnom Penh, Cambodia, 6 Faculty of Medicine, University of Health Sciences, Phnom Penh, Cambodia, 7 Obstetric Department, Calmette Hospital, Phnom Penh, Cambodia, 8 ANRS, University of Health Sciences, Phnom Penh, Cambodia, 9 HIV Unit, Infectious Diseases Department, Geneva University Hospital, Geneva, Switzerland

* sothearamoeung@ymail.com

**Data Availability Statement:** The study data will not be publicly available because they are sensitive health data that could compromise the confidentiality of the patients included. This is in

## Abstract

In Cambodia, traditional medicine was commonly described as being used by pregnant women at two time points: one month before birth and during early postpartum. The present study aims to describe traditional medicine consumption during postpartum phase for women enrolled in the TA PROHM study and to investigate the possible association between traditional medicine consumption and acute liver toxicity. An ethnobotanical survey was conducted in 2 groups of HBV-infected pregnant women (with and without postpartum hepatocellular injury) enrolled in the study. Hepatocellular injury was defined by having Alanine Aminotransferase (ALT) > 2.5 times the Upper Limit of Normal (ULN = 40 U/L) at the 6th week postpartum visit. Interviews were done using a standardized questionnaire. Plant samples were collected and later identified by two traditional healers. Chi-square test was used to find the association between hepatocellular injury and traditional medicine consumption or a specific plant species. In total, 75 women were enrolled and 52 (69.3%) used at least one traditional remedy composed of 123 different plants and 12 alcoholic macerations of porcupine stomach. Orally consuming at least one remedy with alcohol was significantly associated with hepatocellular injury (33% *vs* 13%, p = 0.034). Among the 123 plants species identified, four were found to be associated with hepatocellular injury, namely *Amphineurion marginatum* (Roxb.) D.J.Middleton [Apocynaceae] (p = 0.022), *Selaginella tamariscina* (P.Beauv.) Spring [Selaginellaceae] (p = 0.048), *Mitragyna speciosa* Korth. [Rubiaceae] (p = 0.099) and *Tetracera indica* (Christm. & Panz.) Merr. [Dilleniaceae] (p = 0.079). Consumption of traditional medicine in postpartum is a common practice for women enrolled in the TA PROHM study. Alcohol-based remedies may exacerbate the risk of acute

accordance with French and European legislation on clinical research and personal health data. Nevertheless, the research dataset that underlie the results reported in this Article including deidentified participant data and data dictionary can be made available through a request to the Scientific Advisory Board of the TA PROHM study and to the French National Agency for Research on AIDS and Viral Hepatitis. The access request, if validated, will be framed by an agreement between the sponsor and the applicant. Additional documents, including the questionnaires and the informed consent forms can also be made available. The dataset and the additional documents will be available immediately after publication and could be shared with researchers who provide a methodologically sound proposal and after agreement from the Scientific Advisory Board and the French National Agency for Research on AIDS and Viral Hepatitis. Proposals should be submitted to Ventzislava Petrov-Sanchez (Email: ventzislava.petrov-sanchez@inserm.fr), who is the responsible of the clinical research department of the French National Agency for Research on AIDS and Viral Hepatitis.

**Funding:** This work was supported by the French National Agency for Research on AIDS, Viral Hepatitis and Emerging infectious diseases. The funders had no role in study design, data collection and analysis, decision to publish, or preparation of the manuscript.

**Competing interests:** The authors have declared that no competing interests exist.

**Abbreviations:** ALT, Alanine aminotransferase; ANRS, Agence nationale de recherches sur le sida et les hépatites virales et les maladies infectieuses émergeantes; AST, Aspartate aminotransferase; DILI, Drug-induced liver injury; HAV, Hepatitis A virus; HBV, Hepatitis B virus; HBsAg, Hepatitis B virus antigen; HCV, Hepatitis C virus; HEV, Hepatitis E virus; HILI, Herb-induced liver injury; HIV, Human immunodeficiency virus; IFCC, International Federation of Clinical Chemistry; IQR, Interquartile ranges; MTCT, Mother-to-child-transmission; NECHR, National Ethics Committee for Health Research; RDT, Rapid Diagnostic Test; TM, Traditional Medicine; TDF, Tenofovir Disoproxil Fumarate; ULN, Upper Limit of Normal.

hepatocellular injury in HBV-infected women already exposed to immune restoration. The complex mixtures of herbs need to be further evaluated by in vitro and in vivo studies.

## Introduction

Among the estimated 257 millions of people living with Hepatitis B virus (HBV) infection, 45% are in the Western Pacific Region, which includes Cambodia [1]. The majority of the HBV epidemic is driven by perinatal transmission–infection transmitted from mother-to-child during the third trimester of pregnancy *in utero* or at delivery [2]. In 2017, in Cambodia, the prevalence of HBV infection among pregnant women was estimated to be 4.4% [3]. From 2017 to 2020, a multicenter interventional prospective study, the ANRS 12345 TA PROHM study, was conducted to assess the effectiveness of an immunoglobulin (HBIg)-free strategy to prevent HBV Mother-to-Child-Transmission (MTCT) in Cambodia. In this study, all women enrolled were evaluated for liver disease at the 6$^{th}$ week postpartum. During this evaluation, several women experienced elevated liver transaminase level and, after exclusion of other potential causes (alcohol consumption, medication, HIV, HCV, HAV and HEV infections), the immune restoration and the use of traditional medicine (TM) in early postpartum phase were identified as possible explanations.

Historically, Khmer medicine is based on the use of natural substances (e.g., herbs, animals, minerals) and traditional practices (e.g., dermabrasive practices, spiritual rites) [4]. Nowadays, it continues to play a significant role in the daily life of Cambodian people. During pregnancy, traditional medicine is commonly described as being used at two time points: one month before birth to facilitate the delivery and during early postpartum phase to increase the breast milk production and prevent "toas" (multi-factorial physical and psychological postpartum disorders) [5]. Herbal treatments used to treat or prevent postpartum disorders are sometimes prepared in alcohol, and thus could present a risk of liver toxicity. Furthermore, herbal medicines are also used for the treatment of liver diseases. Some of these plants are reported to be hepatotoxic [6]. Drug-induced liver injury (DILI) following herbal medicine consumption have already been reported in China [7] and Korea [8]. Adverse effects of Cambodian traditional medicine used during pregnancy and postpartum are poorly described, specifically in HBV-infected pregnant women at risk of liver disease.

The present study aims to describe traditional medicine consumption during postpartum phase for women enrolled in the TA PROHM study in terms of ingredients used (plants, animals, minerals, etc.), and method of preparation and administration, as well as to investigate the possible association between traditional medicine consumption and acute liver toxicity.

## Materials and methods

This study is a retrospective ethnobotanical survey nested in the ANRS 12345 TA PROHM study (ClinicalTrials.gov identifier: NCT02937779). The ANRS TA PROHM study is a multicenter observational and interventional prospective study conducted in five maternities in Cambodia from October 2017 to October 2019 aiming to assess the effectiveness of a HBIg-free strategy to prevent HBV MTCT based on 1/ the use of HBsAg and HBeAg rapid diagnostics tests (RDTs) and ALT level serial algorithm to identify women at risk of HBV MTCT 2/ a preventive treatment by tenofovir disoproxil fumarate (TDF) from 24 weeks of pregnancy for eligible pregnant women 3/ an early vaccination at birth (< 2 hours of life) for all newborns. Eligibility criteria for ANRS 12345 TA PROHM study were pregnant women aged 18 or over

on the day of inclusion, tested positive for HBsAg and having signed the study informed consents. Liver disease assessment was performed at the 6th week postpartum for all women enrolled, including liver-function tests.

## Study population

Two groups of women (with and without post-partum hepatocellular injury) were selected from the ANRS 12345 TA PROHM study's database. Hepatocellular injury was defined by having Alanine Aminotransferase (ALT) > 2.5 times the Upper Limit of Normal (ULN = 40 U/L). The first group consisted of women included in the ANRS 12345 TA PROHM study with normal ALT at inclusion visit but with ALT > 2.5 times the Upper Limit of Normal (ULN = 40 U/L) at the 6th week postpartum visit without common risk factors of hepatotoxicity (chronic alcohol consumption, use of drugs with known hepatotoxicity potential, HIV, HCV, HAV and HEV infections). The other group (control group) consisted of women with normal ALT both at inclusion visit in the ANRS 12345 TA PROHM study and at the 6th week postpartum visit. The participants of the second group were randomly selected from the ANRS 12345 TA PROHM study's database.

## Ethical consideration

The study was approved by the National Ethics Committee for Health Research (NECHR) of Cambodia (N˚ 107, April 29, 2019) and the scientific advisory board of the TA PROHM study. All participants signed a written informed consent before entering the TA PROHM study. The participants who were selected for this ethnobotanical study signed another written informed consent before the interview started. The sponsor (French National Agency for Research on AIDS, Viral Hepatitis and Emerging Infectious Diseases, ANRS-MIE) has approved the study and the decision to submit the study for publication.

## Data collection

After the informed consent was obtained, each woman was interviewed face-to-face in Khmer language, preferably at their house by a previously trained interviewer using a standardized questionnaire. When in-house interview was not possible, the interview was conducted by phone. The liver transaminase level status of each woman was blinded to the person carrying out the interview.

The questionnaire was developed based on the results of previous ethnobotanical studies carried out in Cambodia [6, 9], and modified to meet the objectives of the present study. It was then tested on 8 women to validate its use. The questionnaire consisted of two parts. The first part detailed sociodemographic information, the traditional remedies consumed and their period of consumption (during pregnancy or postpartum), as well as the western-medicine consumption. Then, for each type of traditional remedy (e.g., preparations containing one plant or a package of plants, alcoholic drink, pill or cream . . .), information on duration, posology, ingredients used, method of preparation (e.g., decoction, alcoholic maceration . . .) and administration (e.g., oral, local, inhalation, . . .), as well as reasons for use and source of information was collected. The information related to a remedy could also be given by the person who delivered it (e.g., members of the family or traditional practitioners). The ingredients of each herbal remedy (e.g., package of plants from market or single herb from home garden) were collected at the time of the interview (if the woman had kept them) and stored until identification; otherwise, if possible, those ingredients were collected later at the place/market where they originated from.

### Botanical identification

The identification of herbal ingredients was performed as follow: First, packages of dried plants were labeled with the patient code and a remedy number and stored in Phnom Penh at the Institut Pasteur du Cambodge and at the Phytochemistry laboratory of the University of Health Sciences. The plant parts of each package were then sorted by similar appearance, packed in separate small plastic bags and labelled with the patient code, remedy number and sub-number (Fig 1). Two recognized traditional healers, members of the National Center of Traditional Medicine in Phnom Penh, identified the local name of the plant specimen based on their knowledge in medicinal plants. In case of discordance between them, a consensus was obtained after a discussion with both of them. When an association between a plant species and hepatocellular injury was reported, a third traditional healer was involved for the identification of the plant. Latin names were obtained using the book "Cambodian Plants", volume 1–5, published by the National center of Traditional Medicine (Ministry of Health in Cambodia) and cross-checked for accuracy by consulting The World Flora Online website (http://www.worldfloraonline.org/). For 16 women, the local name of homegrown herbs was given by the women directly during the interview, then the scientific name was obtained using the books mentioned above.

### Biological analysis

In the TA PROHM study, HBsAg and HBeAg RDTs were performed following a sequential testing algorithm. First, all pregnant women were tested with the SD BIOLINE HBsAg RDT

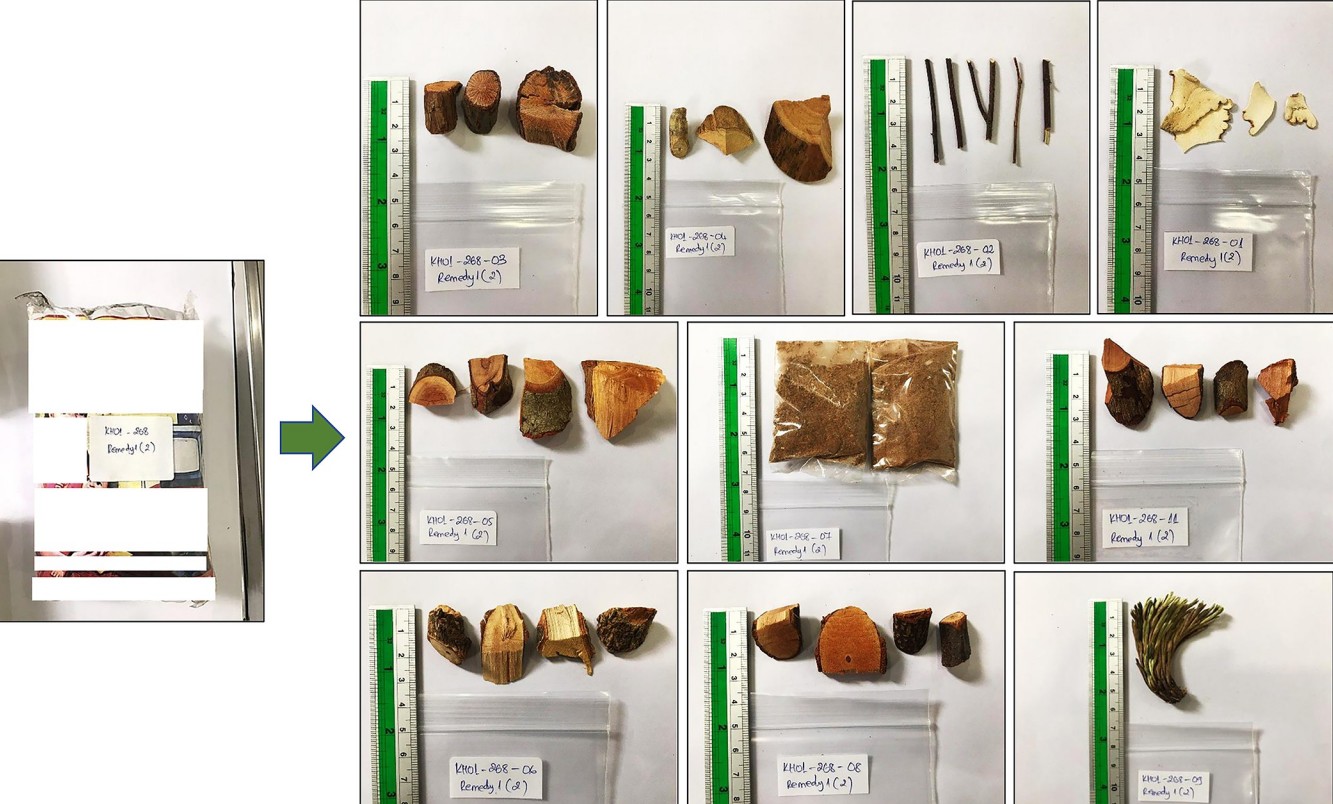

**Fig 1. Separation of plant parts for identification.**

(Standard Diagnostics [SD], INC., Kyonggi-do, Korea). Women positive for HBsAg RDT were further tested with the SD BIOLINE HBeAg RDT (Standard Diagnostics [SD], INC., Kyonggi-do, Korea) to assess eligibility for TDF antiviral prophylaxis.

ALT level was measured on ABX PENTRA C400 by IFCC (International Federation of Clinical Chemistry) method (UV without pyridoxal phosphate) at the Institut Pasteur du Cambodge.

### Statistical method

In absence of possible assumption about the traditional medicine exposure rates in pregnant women in Cambodia, a formal sample size calculation was not performed. A pragmatic approach was used by reaching the maximum of women having experienced a hepatocellular injury in postpartum during the study.

Participants' characteristics are presented as percentages, medians, interquartile ranges (IQR). Comparison between the two groups of women (i.e. with and without hepatocellular injury) was performed using Student's t-test for quantitative variables and Chi-square test for qualitative variables. The proportion of women using a specific plant species were compared between the two groups using Chi-square test. When an association between a plant species and hepatocellular injury was found (p<0.1 regarding the low sample size), a specific literature review was performed to search for its previously described therapeutic and/or pharmacological effects and its potentially known liver toxicity. All the statistical analyses were performed using STATA 11 software.

### Results

From March 12th 2019 to April 7th 2020, 108 women were contacted, of whom 25 could not be reached, one refused to participate and 7 were not eligible (because their ALT level at the 6th week postpartum ranged between the ULN and 2.5 times the ULN). Finally, 75 women were included in the analysis, of whom 36 had hepatocellular injury and 39 had normal ALT level at the 6th week postpartum.

The characteristics of the participants at the time of the interview were reported in Table 1. There was no significant difference between the two groups except for AST and ALT level at the 6th week postpartum.

A total of 151 different traditional remedies were collected, composed of 123 different plants and 12 alcoholic macerations of porcupine stomach. Overall, 80 traditional remedies were ingested, 39 applied on the skin, 3 inhaled and 25 used for steamed bath and 4 were not reported. Some samples of different types of traditional remedies collected are shown in Fig 2. The traditional medicine was started in a median delay of 11 days (IQR, 7–30) after delivery. The median duration of the use of traditional medicine was 15 days (IQR, 5–30). At the time of the interview, no women reported to have used additional conventional medication during postpartum.

The mode of administration and preparation are reported in Table 2. Overall, 52 women (69.3%) used at least one traditional remedy regardless of the mode of administration, 29/36 (80.6%) in the hepatocellular injury group as compared to 23/39 (59.0%) in the no hepatocellular injury group (p = 0.037). In total, 17 women (22.7%) orally consumed traditional remedies with alcohol, 12/36 (33.3%) in the hepatocellular group as compared to 5/39 (12.8%) in the no hepatocellular group (p = 0.034). The median ALT level was 148.5 U/L (IQR, 114.5–204.5) for the 12 women taking remedies in alcohol as compared to 126 U/L (IQR, 102–149) for the 15 women taking remedies without alcohol (p = 0.16).

**Table 1. Characteristics of women included in the study (N = 75).**

| Description | Hepatocellular injury N = 36 | No Hepatocellular injury N = 39 | p Value |
|---|---|---|---|
| **Age** (in years, median, IQR*) | 31 (28–35) | 32 (28–36) | 0.656 |
| **Location**, N (%) | | | 0.630 |
| *Phnom Penh* | 12 (33.3) | 11 (28.2) | - |
| *Province* | 24 (66.7) | 28 (71.8) | - |
| **Ethnic group**, N (%) | | | 1 |
| *Khmer* | 36 (100) | 39 (100) | - |
| **Level of education**, N (%) | | | 0.926 |
| *No education or primary* | 13 (36.1) | 13 (33.3) | - |
| *Secondary* | 15 (41.7) | 18 (46.2) | - |
| *University* | 8 (22.2) | 8 (20.5) | - |
| **Average family income** per month, N (%) | | | 0.477 |
| *< 100 USD* | 2 (5.6) | 1 (2.6) | - |
| *100–300 USD* | 13 (36.1) | 16 (41.0) | - |
| *300–600 USD* | 9 (25.0) | 15 (38.4) | - |
| *600–1000 USD* | 4 (11.1) | 3 (7.7) | - |
| *>1000 USD* | 8 (22.2) | 4 (10.3) | - |
| **Medical status** | | | |
| *Positive HBeAg status, N (%)* | 12 (33.3) | 11 (28.2) | 0.630 |
| *Tenofovir (TDF) treatment, N (%)* | 15 (41.7) | 14 (35.9) | 0.391 |
| *Liver biological test* | | | |
| *AST at week 6 post-partum, median (IQR)* | 110 (80–123) | 27 (24–31) | < 0.001 |
| *ALT at week 6 post-partum, median (IQR)* | 136 (110–162) | 28 (22–33) | < 0.001 |

Among all plants collected and identified (S1 Table), four were significantly associated with hepatocellular injury (p<0.1), namely *Amphineurion marginatum* (Roxb.) D.J.Middleton [Apocynaceae], *Mitragyna speciosa* Korth. [Rubiaceae], *Tetracera indica* (Christm. & Panz.) Merr. [Dilleniaceae] and *Selaginella tamariscina* (P.Beauv.) Spring [Selaginellaceae] (Table 3).

## Discussion

Traditional medicine consumption was a common practice during postpartum period among women enrolled in the TA PROHM study and included in this ancillary study. Orally consuming at least one traditional remedy with alcohol was significantly associated with post-partum hepatocellular injury. Among the 123 plants identified in the collected traditional remedies, two plants known for containing hepatotoxic compounds (*Amphineurion marginatum* and *Mitragyna speciosa*) [10, 11] were reported as possibly associated with hepatocellular injury [10–12].

In total, 69.3% of the women consumed traditional remedies in our study, composed of 123 different plants. Infusions, decoctions of herbal ingredients (dry or fresh plants) and/or alcoholic macerations (dry plants or porcupine stomach) were reported for 62.7% of women orally absorbing traditional remedies. In Cambodia, the use of herbal medicine during postpartum is a common practice. A cross-sectional survey conducted in outpatients in two urban and two rural primary health centres reported that 27% of participants had at least one consultation with a traditional healer during the year preceding the survey, and herbal medicine represented 89% of the prescriptions [13]. In 2 studies conducted in Malaysia, herbal medicine was used by 52.9% [14] and 58.9% [15] of the participants respectively during postpartum phase. The common method of preparation is to boil the herbal ingredients or, optionally, to

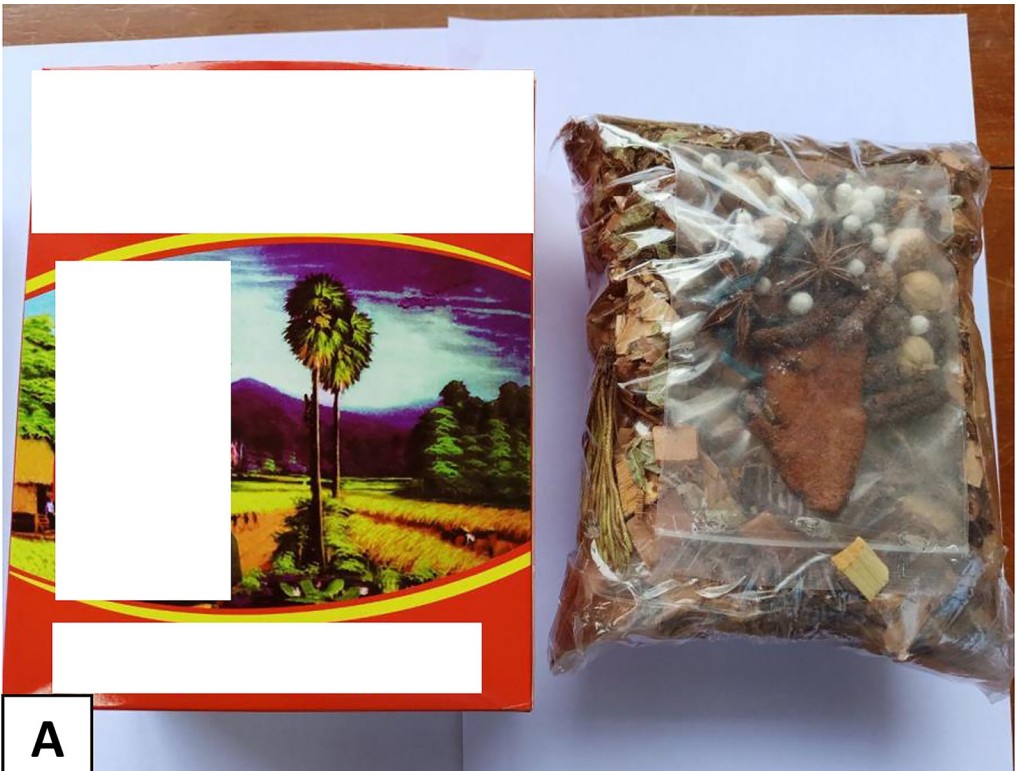
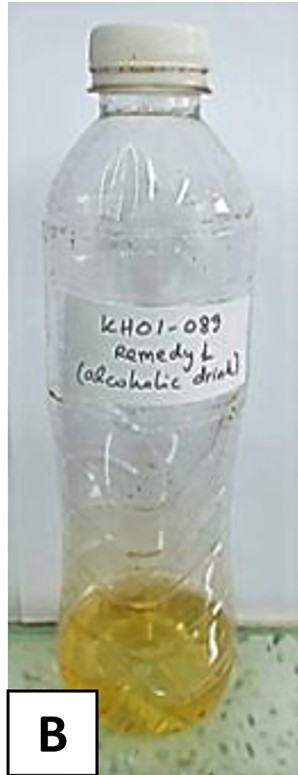
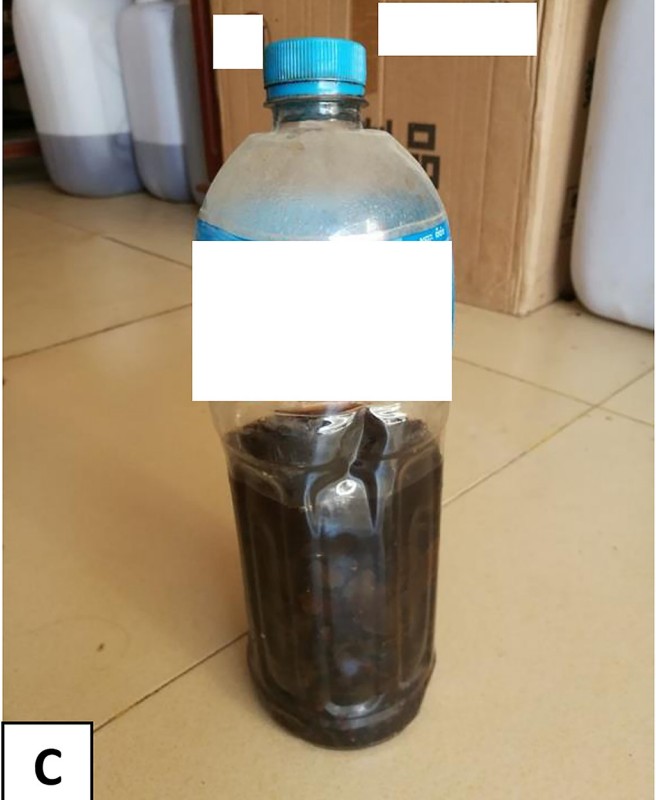
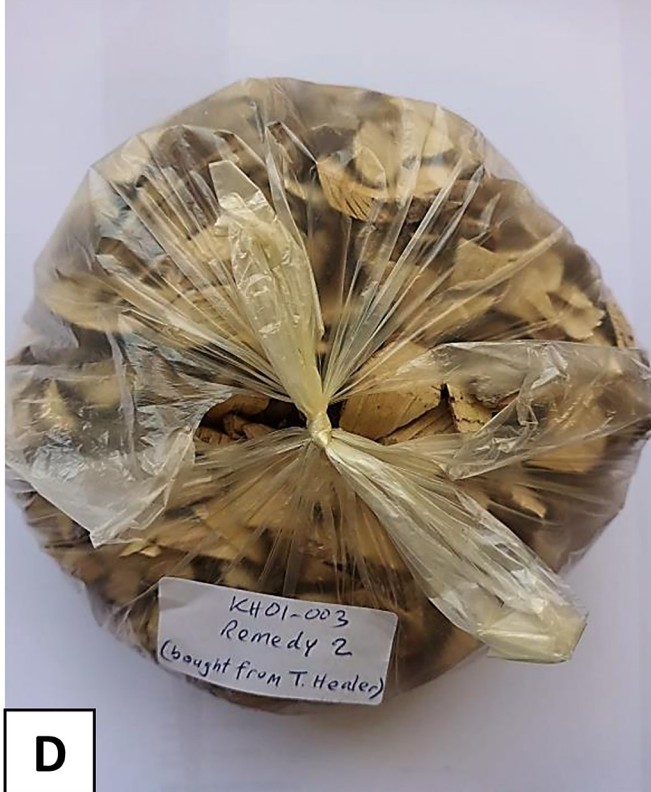

**Fig 2. Different types of traditional remedies used by the participants.** (A) Mix of dried plants sold at a market in Phnom Penh. (B) Alcoholic maceration of porcupine stomach. (C) Juice of crushed fresh plants. (D) Mix of dried plants obtained from a traditional healer.

**Table 2. Modes of administration and preparation (n = 75).**

| | Hepatocellular injury N = 36 | No Hepatocellular injury N = 39 | Total N = 75 | p Value |
|---|---|---|---|---|
| *Used at least one traditional remedy* | 29 (80.6) | 23 (59.0) | 52 (69.3) | 0.037 |
| **Mode of administration, N (%)** | | | | |
| *Orally consumed at least 1 remedy* | 27 (75.0) | 20 (51.3) | 47 (62.7) | 0.034 |
| *Applied at least 1 remedy* | 14 (38.9) | 15 (38.5) | 29 (38.7) | 0.970 |
| *Used at least 1 remedy as steam bath* | 22 (61.1) | 26 (66.7) | 48 (64.0) | 0.617 |
| **Mode of preparation, N (%)** | | | | |
| *Orally consumed at least 1 remedy with alcohol* | 12 (33.3) | 5 (12.8) | 17 (22.7) | 0.034 |
| *Took at least 1 remedy containing porcupine stomach** | 9 (25.0) | 2 (5.1) | 11 (14.7) | 0.015 |
| *Orally consumed at least 1 remedy without alcohol* | 15 (41.7) | 15 (38.5) | 30 (40) | 0.77 |
| *Inhaled at least 1 remedy or applied remedy followed by steam bath* | 14 (38.9) | 15 (38.5) | 29 (38.7) | 0.970 |
| **Number of remedies ingested by each patient (median, IQR)** | 2 (1–3) | 1 (0–3) | 2 (0–3) | 0.105 |

*Porcupine stomach is always associated with alcohol.

macerate the plant mixture in rice wine, then to drink it [16, 17]. The use of porcupine stomach is also a common practice in postpartum period and has been mentioned in some studies [9, 18]. In Cambodia, there is scarce information related to the composition of plant mixture orally consumed during postpartum. To date, 65 and 75 plant species consumed during postpartum as an infusion tea or alcoholic maceration have been identified in Prey Lang area and Mondulkiri province respectively [9, 19]. A review of medicinal plants for women's healthcare in Southeast Asia focusing on about 200 studies reported that nearly 2000 different plant species are used in over 5000 combinations. These plants are mainly used to treat postpartum hemorrhage, postpartum pain, microbial infections, and as lactation stimulants [20].

After eliminating common risk factors of hepatotoxicity, two possible causes of postpartum hepatocellular injury were identified: a biochemical-hepatic flare and the consumption of traditional medicine, especially if it is prepared in the form of alcoholic maceration. Biochemical-hepatic flares with increased ALT levels have been reported during the first months of nucleoside analogues initiation and in early postpartum when immune reconstitution occurs [21, 22]. The reported rates of ALT flares in postpartum are variable according to the definition and the presence of antiviral therapy and could be more common for positive HBeAg women [23]. In our study, the proportions of positive HBeAg women and TDF-treated women are similar in the two groups. TDF was initiated from 24 weeks of amenorrhea (so 4 to 5 months before the ALT assessment), which is a long delay for a treatment flare. Orally consuming at least one traditional remedy prepared in alcohol in postpartum period was significantly associated with acute hepatocellular injury. Drug-induced liver injury (DILI) during pregnancy and postpartum was documented with many medications but few with herbal medications and traditional medicines [24]. In a recent systematic review of 31 articles evaluating DILI/Herb-induced liver injury (HILI) in general population, 60.7% were reported by conventional drugs

**Table 3. Plants significantly associated with hepatocellular injury (N = 75).**

| Plant name | Total N(%) | Hepatocellular injury N(%) | No Hepatocellular injury N(%) | p Value |
|---|---|---|---|---|
| *Amphineurion marginatum* (Roxb.) D.J.Middleton [Apocynaceae] | 5 (10.6) | 5 (13.9) | 0 (0.0) | 0.022 |
| *Selaginella tamariscina* (P.Beauv.) Spring [Selaginellaceae] | 4 (8.5) | 4 (11.1) | 0 (0.0) | 0.048 |
| *Mitragyna speciosa* Korth. [Rubiaceae] | 6 (12.8) | 5 (13.9) | 1 (2.6) | 0.099 |
| *Tetracera indica* (Christm. & Panz.) Merr [Dilleniaceae] | 9 (19.1) | 7 (19.4) | 2 (5.1) | 0.079 |

and 25.0% by herbs [25]. In this systematic review, hepatocellular injury was significantly more frequent for HILI as compared to DILI and cholestatic injury more frequent for DILI as compared to HILI. However, in this review, while 35 different herbs were identified, no information was provided about a possible association with alcohol. Alcoholic maceration of plants or porcupine stomach, as reported in our study, may represent an isolated or a cumulative and exacerbating factor of acute hepatocellular injury for patients with pre-existing risk factor such as HBV infection and exposed to immune restoration. Further investigations are needed to understand whether alcohol alone or the preparation is responsible for this toxicity.

Four of the 123 plants identified were associated with post-partum hepatocellular injury, and two, *Amphineurion marginatum* and *Mitragyna speciosa*, have been reported to contain hepatotoxic compounds. Root, leaf and stem material of *Amphineurion marginatum* contain pyrrolizidine alkaloids [10]. This group of compounds are found in a variety of plant species and were the cause of large and recurring episodes of acute hepatotoxicity in several countries [12]. Following oral absorption, these alkaloids are converted to their ester metabolites, which are biological alkylating agents capable of causing tissue damage and inducing genetic mutations. In human, during acute poisoning, these alkaloids cause thickening and fibrosis of the hepatic sinusoids and central veins, which leads to increased pressure in the portal venous system; collectively termed "hepatic sinusoidal obstruction syndrome or veno-occlusive disease". However, the time span between the onset of the consumption and the apparition of signs of intoxication was not mentioned [12, 26]. *Mitragyna speciosa*, on the other hand, has been found to be hepatotoxic in *in vitro* and *in vivo* studies [27, 28] as well as in a number of case reports [29, 30]. Its hepatotoxicity is probably due to its main alkaloid, mitragynine, which has been found to cause liver injury via hepatocellular damage and cholestatic mechanisms [11].

Our study has some limitations. First, our results are from a small sample of HBV-infected women and should therefore be confirmed with a larger sample size and could not be generalized to HBV-uninfected pregnant women. Secondly, in the absence of HBV DNA quantification at week-6 postpartum, we cannot evaluate an HBV reactivation related to an immune restoration. Thirdly, daily and total quantity of alcohol contained in alcohol-based remedies could not be determined due to heterogeneous and unclear posology. Lastly, the study should have benefited from an *in vitro* analysis of cytolysis on hepatocytes to confirm the possible toxicity of the four plants identified.

In conclusion, the consumption of traditional medicine in postpartum is a common practice among the women enrolled in the TA PROHM study. Alcohol-based remedies may exacerbate the risk of acute hepatocellular injury in HBV-infected women already exposed to immune restoration. The complex mixture of herbs, which have a higher metabolic demand for liver and potential interaction among ingredients, needs to be further evaluated by *in vitro* and *in vivo* studies.

## Supporting information

**S1 Table. List of plants identified in the study.**
(PDF)

**S1 File. Completed PLOS' questionnaire on inclusivity in global research.**
(DOCX)

**S2 File. Synopsis of the ANRS 12345 TA PROHM study.**
(PDF)

## Acknowledgments

The authors would like to thank all pregnant women and their families for their participation in the study; Ms SOENG Chivly for her help in samples storage at laboratory, the traditional healers for identification of the plants, namely Mr. KY Bouhaing, Mr. HUON Chhom and Mr. THAY Savy; the laboratory staff of Institut Pasteur du Cambodge for liver function test realization; all the members of the Scientific Committee of the TA PROHM study; Claire Rekacewicz, Isabelle Fournier-Nicolle, Nicolas Rouveau, Maria-Camila Calvo Cortes and Laura Fernandez for their support.

## Author Contributions

**Conceptualization:** François Chassagne, Laurence Borand, Sothea Kim, Olivier Segeral.

**Data curation:** Sophie Goyet, Sovann Nhoeung, Socheat Ly.

**Formal analysis:** Sophie Goyet.

**Investigation:** Sotheara Moeung, François Chassagne, Lynecta Sun, Dorina Yang, Steve Vilhem, Bunnet Dim, Linda Sov, Vouchleang Sreng, Sokda Chorn.

**Methodology:** Sotheara Moeung, François Chassagne, Sophie Goyet, Olivier Segeral.

**Project administration:** Samsorphea Chhun, Laurence Borand, Sothea Kim, Olivier Segeral.

**Writing – original draft:** Sotheara Moeung, François Chassagne, Olivier Segeral.

**Writing – review & editing:** Sotheara Moeung, François Chassagne, Sophie Goyet, Laurence Borand, Olivier Segeral.

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
