## [Decision Letter · Decision Letter 0]

2 Mar 2023

PONE-D-22-34415Traditional medicine consumption in postpartum for HBV-infected women enrolled in the ANRS 12345 TA PROHM study in CambodiaPLOS ONE

Dear Dr. MOEUNG,

Thank you for submitting your manuscript to PLOS ONE. After careful consideration, we feel that it has merit but does not fully meet PLOS ONE’s publication criteria as it currently stands. Therefore, we invite you to submit a revised version of the manuscript that addresses the points raised during the review process.

We look forward to receiving your revised manuscript.

Kind regards,

Marcello Iriti, Ph.D.

Academic Editor

PLOS ONE

Journal Requirements:

6. We note that Figures 1 and 2 in your submission contain copyrighted images. All PLOS content is published under the Creative Commons Attribution License (CC BY 4.0), which means that the manuscript, images, and Supporting Information files will be freely available online, and any third party is permitted to access, download, copy, distribute, and use these materials in any way, even commercially, with proper attribution. For more information, see our copyright guidelines: http://journals.plos.org/plosone/s/licenses-and-copyright.

a. You may seek permission from the original copyright holder of Figures 1 and 2 to publish the content specifically under the CC BY 4.0 license. 

Additional Editor Comments:

Authors should mitigate their findings and modify conclusions, as requested by the reviewer 2.

Reviewers' comments:

Reviewer's Responses to Questions

**Comments to the Author**

1. Is the manuscript technically sound, and do the data support the conclusions?

Reviewer #1: Yes

Reviewer #2: No

2. Has the statistical analysis been performed appropriately and rigorously? 

Reviewer #1: Yes

Reviewer #2: No

3. Have the authors made all data underlying the findings in their manuscript fully available?

Reviewer #1: No

Reviewer #2: Yes

4. Is the manuscript presented in an intelligible fashion and written in standard English?

Reviewer #1: Yes

Reviewer #2: No

5. Review Comments to the Author

Reviewer #1: This is good work in an area not commonly studied by many researchers and yet herbal Medicine is used quite a lot by women during pregnancy. I have checked and noted that the work has not been published elsewhere before and is written in very good English. The authors have however, not compared the findings to studies of similar nature done in other countries or within their own country if any to put this current one to perspective. it would be a good one to add if the authors don't mind.

Reviewer #2: In the present study the authors aim to describe traditional medicine consumption during postpartum phase for women enrolled in the TA PROHM study and to investigate the possible association between traditional medicine consumption and acute liver toxicity.

As pointed by the authors the study has several limitations, additionally the authors conclude that consumption of remedy with alcohol is associated with hepatocellular injury. However it may not be concluded that the hepatocellular injury is associated with remedy, it may be purely associated with the alcohol content rather then the medicinal plant.

6. PLOS authors have the option to publish the peer review history of their article (what does this mean?). If published, this will include your full peer review and any attached files.

Reviewer #1: **Yes: **Joseph Oloro

Reviewer #2: No

---

## [Author Response · Author response to Decision Letter 0]

9 May 2023

We would like to thank all Reviewers for their careful and thorough reading of this manuscript and for the thoughtful comments and constructive suggestions, which help to improve the quality of this manuscript. 

Responses to Reviewer #1:

Reviewer #1: This is good work in an area not commonly studied by many researchers and yet herbal Medicine is used quite a lot by women during pregnancy. I have checked and noted that the work has not been published elsewhere before and is written in very good English. The authors have however, not compared the findings to studies of similar nature done in other countries or within their own country if any to put this current one to perspective. it would be a good one to add if the authors don't mind.

Reply:

We highly appreciate the feedback from the reviewer. We have thoroughly re-performed the literature review. Unfortunately, we haven’t found any studies which assess the toxicity of herbal medicine used during postpartum. However, we have found some articles from other countries mentioning the use of herbal medicine during this particular period. This supports our finding that the postpartum use of herbal medicine is still as common in other places as in Cambodia. We have included the result of the literature review in the Discussion section, page 15, lines 287 and 288 of the revised manuscript.

Responses to Reviewer #2:

Reviewer #2: In the present study the authors aim to describe traditional medicine consumption during postpartum phase for women enrolled in the TA PROHM study and to investigate the possible association between traditional medicine consumption and acute liver toxicity. As pointed by the authors the study has several limitations, additionally the authors conclude that consumption of remedy with alcohol is associated with hepatocellular injury. However it may not be concluded that the hepatocellular injury is associated with remedy, it may be purely associated with the alcohol content rather then the medicinal plant.

Reply:

We completely agree with the reviewer. We have by no means associated hepatocellular injury solely to the plants and indeed, hepatocellular injury could be secondary to the consumption of the alcohol alone. However, as alcohol is an inseparable component of the remedy for some specific plants, it is impossible to conclude. In vitro analysis of cytolysis on hepatocytes would be useful to confirm the possible toxicity of these plants. In our article, the terminology “alcohol-based remedies” as a whole was used and associated with the hepatocellular injury. 

The discussion was completed lines 318 – 321 for that purpose. “Alcoholic maceration of plants or porcupine stomach, as reported in our study, may represent an isolated or a cumulative and exacerbating factor of acute hepatocellular injury for patients with pre-existing risk factor such as HBV infection and exposed to immune restoration. Further investigations are needed to understand whether alcohol alone or the preparation is responsible for this toxicity.”

---

## [Decision Letter · Decision Letter 1]

26 Jun 2023

Traditional medicine consumption in postpartum for HBV-infected women enrolled in the ANRS 12345 TA PROHM study in Cambodia

PONE-D-22-34415R1

Dear Dr. MOEUNG,

We’re pleased to inform you that your manuscript has been judged scientifically suitable for publication and will be formally accepted for publication once it meets all outstanding technical requirements.

Kind regards,

Marcello Iriti, Ph.D.

Academic Editor

PLOS ONE

Additional Editor Comments (optional):

Reviewers' comments:

Reviewer's Responses to Questions

**Comments to the Author**

1. If the authors have adequately addressed your comments raised in a previous round of review and you feel that this manuscript is now acceptable for publication, you may indicate that here to bypass the “Comments to the Author” section, enter your conflict of interest statement in the “Confidential to Editor” section, and submit your "Accept" recommendation.

Reviewer #1: All comments have been addressed

Reviewer #2: All comments have been addressed

2. Is the manuscript technically sound, and do the data support the conclusions?

Reviewer #1: Yes

Reviewer #2: Yes

3. Has the statistical analysis been performed appropriately and rigorously? 

Reviewer #1: Yes

Reviewer #2: Yes

4. Have the authors made all data underlying the findings in their manuscript fully available?

Reviewer #1: Yes

Reviewer #2: Yes

5. Is the manuscript presented in an intelligible fashion and written in standard English?

Reviewer #1: (No Response)

Reviewer #2: Yes

6. Review Comments to the Author

Reviewer #1: The authors have done their best to improve on the paper by getting relevant papers to the what they had researched on for the purpose of comparison. it is better now and can be accepted for publication.

Reviewer #2: The reviewers concerns have been satisfactorily addressed and the manuscript has been updated to reflect the changes.

7. PLOS authors have the option to publish the peer review history of their article (what does this mean?). If published, this will include your full peer review and any attached files.

Reviewer #1: **Yes: **Joseph Oloro

Reviewer #2: No

---

## [Editor Report · Acceptance letter]

31 Jul 2023

PONE-D-22-34415R1 

Traditional medicine consumption in postpartum for HBV-infected women enrolled in the ANRS 12345 TA PROHM study in Cambodia 

Dear Dr. Moeung :

I'm pleased to inform you that your manuscript has been deemed suitable for publication in PLOS ONE. Congratulations! Your manuscript is now with our production department. 

Kind regards, 

on behalf of

Prof. Marcello Iriti 

Academic Editor

PLOS ONE